# Managing Trust and Detecting Malicious Groups in Peer-to-Peer IoT Networks

**DOI:** 10.3390/s21134484

**Published:** 2021-06-30

**Authors:** Alanoud Alhussain, Heba Kurdi, Lina Altoaimy

**Affiliations:** 1Computer Science Department, College of Computer and Information Sciences, King Saud University, Riyadh 11451, Saudi Arabia; aalhussain1@ksu.edu.sa; 2Mechanical Engineering Department, School of Engineering, Massachusetts Institute of Technology (MIT), Cambridge, MA 02142, USA; 3Information Technology Department, College of Computer and Information Sciences, King Saud University, Riyadh 11451, Saudi Arabia; laltoaimy@ksu.edu.sa

**Keywords:** IoT, neural networks, peer-to-peer networks, reputation management, trust management

## Abstract

Peer-to-peer (P2P) networking is becoming prevalent in Internet of Thing (IoT) platforms due to its low-cost low-latency advantages over cloud-based solutions. However, P2P networking suffers from several critical security flaws that expose devices to remote attacks, eavesdropping and credential theft due to malicious peers who actively work to compromise networks. Therefore, trust and reputation management systems are emerging to address this problem. However, most systems struggle to identify new smart models of malicious peers, especially those who cooperate together to harm other peers. This paper proposes an intelligent trust management system, namely, Trutect, to tackle this issue. Trutect exploits the power of neural networks to provide recommendations on the trustworthiness of each peer. The system identifies the specific model of an individual peer, whether good or malicious. The system also detects malicious collectives and their suspicious group members. The experimental results show that compared to rival trust management systems, Trutect raises the success rates of good peers at a significantly lower running time. It is also capable of accurately identifying the peer model.

## 1. Introduction

The booming number of IoT devices is paving the way for peer-to-peer (P2P) architecture to dominate in IoT platforms so that connectivity and latency issues usually associated with centralized cloud services architecture are avoided [1]. Although the cloud model is easy to manage and economical to scale, the requirements for uninterrupted internet connections are too costly and burdensome for many IoT devices. However, P2P architecture supports emerging IoT applications such as proximity sharing, where colocated devices can cooperate with each other in real time to complete a task or share a resource. However, in this case, devices (peers) need to discover each other, trust each other, and then make a connection [2].

To illustrate the important role of trustworthy communication between P2P IoT devices, consider a smart home where IoT devices provide functions that help make users’ daily routines more convenient. A smart door lock is a smart home IoT device that has seen snowballing adoption in recent years. An example of such a device is one that communicates with another device that includes an identity verification mechanism, such as a mobile phone. Accordingly, the smart lock identifies a user as the homeowner and unlocks the home door automatically as they approach. When a homeowner leaves home, the smart lock automatically locks the door behind them. This is an idealized case where both devices are trustworthy and the user has immutable control over them. Problems arise if a malicious actor takes over control of one or more of these devices. This could occur due to any of the common IoT devices’ vulnerabilities, such as poor configuration or using default passwords. For instance, if a malicious actor were to compromise a user identification mechanism, the smart lock can give them control over who comes in or out of a home, thus enabling intruders to have access to the house and lock residents out of it [3]. In this context, the user identity is the resource, the smart phone is the resource provider while the smart lock is the recipient. After each locking and unlocking event, the smart lock would send a confirmation message to the user to ask if they are aware of the door locking/unlocking event. The feedback about the transaction is sourced from multiple channels (i.e., phone, email, etc.); this limits the impact of bad transactions.

Peers in a P2P network vary greatly in their behavior, from newcomers to pretrusted, good and malicious peers. Newcomers are peers who have just joined the network. Their behaviors are revealed gradually once they start communicating with other peers. Pretrusted peers are usually assigned by the founders of the network; thus, they always provide trusted resources and honest feedback on the quality of the resources they receive. Good peers normally provide good resources unless fooled by malicious peers, and they always give honest feedback [4]. However, malicious peers use the vulnerability of P2P networks to distribute bad resources [5,6] and dishonest feedback [7]. Malicious peers are usually further classified into four categories. First, pure malicious peers provide bad resources and dishonest feedback on other peers [8,9]. Second, camouflaged malicious (disguised malicious) peers are inconsistent in their behavior. They normally provide bad resources, but they also offer good resources to enhance their rating and delude other peers [10]. In addition, they always give negative feedback on other peers [11]. Third, feedback-skewing peers (malicious spies) provide good resources but lie in their feedback on other peers [12]. Fourth, malignant providers provide bad resources to other peers. However, they do not lie in their feedback. Malicious peers work individually or assemble in groups [13,14,15]. Malicious behavior can be categorized according to two strategies: isolated and collective. Isolated malicious peers act independently from each other while collective malicious peers cooperate in groups to harm other peers. In contrast, they never harm their group members with bad resources or negative feedback.

Trust management is about developing strategies for establishing dependable interactions between peers and predicting the likelihood of a peer behaving honestly. It focuses on how to create a trustworthy system from trustworthy distributed components [16,17], which is paramount, especially for newly emerging forms of distributed systems [16], such as cloud computing [18,19], fog computing [20] and IoT [1,6]. Trust management systems monitor peers’ conduct and allow them to give feedback on their previous transactions; thus, good and bad peers can be identified. This binary classification of peers is the approach followed by most previous studies [12,21,22,23], disregarding the fact that peers vary greatly in their behavior between the two extremes. Additionally, existing trust management systems considered neither predicting the specific peer model nor identifying other members within his/her malicious group. To bridge this gap, this paper proposes Trutect, which is a trust management system that uses the power of neural networks to detect malicious peers and identify their specific model and other group members, if any. The proposed system is thoroughly evaluated and benchmarked based on rival trust management systems, including EigenTrust [24] and InterTrust [25].

Among the main contributions of this paper are the following:A trust management system that exploits the power of neural networks to identify the specific peer model and its group members.A large dataset of behavioral models in P2P networks that can be used to advance research in the field.A well-controlled evaluation framework to study the performance of a trust management system.

## 2. Literature Review

Due to the rapid adoption of IoT technologies, they are becoming an attractive target for cyber criminals who take advantage of lack of security functions and abilities to compromise IoT devices. Therefore, a plethora of research is emerging to elucidate the security attacks and different security and trust management mechanisms involved in IoT applications [26,27].

In identifying malicious peers, trust and reputation management systems follow different approaches [25], such as trust vectors [24], subjective logic [25] and machine learning [8]. EigenTrust [24] is one of the most commonly used trust management algorithms based on a normalized trust vector. Despite the widespread use of EigenTrust, it suffers from drawbacks, mainly related to the ability of a peer to easily manipulate the ratings it provides on other peers. Additionally, its heavy reliance on pretrusted peers makes them focal points of failure. In [28], an enhancement of the EigenTrust algorithm to ensure that a peer cannot manipulate its recommendation is proposed; hence, this approach is called nonmanipulatable EigenTrust. In [29], another reputation scheme based on EigenTrust where the requester selects a provider peer using the roulette wheel selection algorithm to reduce the reliance of pretrusted peers was introduced. For the same reason, [30] introduced the concept of honest peers, i.e., a peer with a high reputation value who can be targeted, instead of pretrusted peers, by new peers.

Among the early trust management systems is the trust network algorithm using subjective logic (TNA-SL) [31]. Trust in TNA-SL is calculated as the opinion of a peer on another peer based on four components: belief, disbelief, uncertainty and a base rate. Opinions provide accurate trust information about peers; however, the main drawback of TNA-SL is its exponential running time complexity due to the lengthy matrix chain multiplication process for transitive trust calculations. Thus, InterTrust [25] was proposed to overcome this issue. InterTrust maintains the advantages of the original TNA-SL while being scalable and lightweight with low computational overhead by using more scalable data structures and reducing the need for matrix chain multiplications.

Machine learning can help with the prediction and classification of peers’ trustworthiness in either a static or a dynamic manner [32]. In static approaches, the peers’ extracted features are selected offline to produce one final model, thereby decreasing the computational overhead. In contrast, dynamic approaches extract peers’ features online while the application is running, which incrementally updates the model based on past data and offers the ability to detect malicious peers based on new behavioral information. However, these advantages usually result in high computational overheads.

A static generic machine learning-based trust framework for open systems using linear discriminant analysis (LDA) [33] and decision trees (DTs) [34] was presented in [35]. In this approach, an agent’s past behavior is not considered when determining whether to interact with the agent; instead, the false positive rate, false negative rate, and overall falseness are used. A trust management system for the IoT based on machine learning and the elastic slide window technique was proposed in [6]. The main goal was to identify on-off attackers based on static analysis.

Another static approach using machine learning for the problem of trust prediction in social networks was presented in [7]. The study uses recommender systems to predict the trustworthiness of each peer. In [36], a trust-based recommendation system that assesses the trustworthiness of a friend recommendation while preserving users’ privacy in an online social network is introduced. Friends’ features were statically analyzed, and recommendations about peer trustworthiness were derived using the K-nearest neighbors algorithm (KNN).

The work in [37] enables the prediction results to be integrated with an existing trust model. It was applied on an online web service that helps customers book hotels. Features related to the application domain were selected and statically analyzed by several supervised algorithms, such as experience-based Bayesian, regression, and decision tree algorithms. A reputation system for P2P using a Support vector machines algorithm (SVM) was built in [22]. The system dynamically collects information on the number of good transactions in each single time slot. The system outperformed the other system even more at very high imbalance ratios because the overall accuracy increased, and the system tended to classify almost all nodes as malicious.

All previously mentioned works are static and focus on the binary classification of peers’ trustworthiness, which means that a peer is either good or bad. In contrast, D-Trust [38] presents a dynamic multilevel social recommendation approach. D-Trust creates a trust-user-item network topology based on dynamic user rating scores. This topology uses a deep neural network, focuses on positive links and eliminates negative links. Another dynamic neural network-based multilevel reputation model for distributed systems was proposed by [15]. It dynamically analyses global reputation values to find the peer with the highest reputation. In [39] a deep neural network was used to build trustworthy communications in Vehicular Ad hoc Networks (VANETs). The trust model evaluates neighbours’ behaviour while forwarding routing information using a software-defined trust-based dueling deep reinforcement learning approach. Additionally, a multilevel trust management framework was proposed in [14]. This approach uses dynamic analysis with an SVM to classify interactions into trustworthy, neutrally trusted, or untrustworthy interactions.

Based on the above surveyed work, trust and reputation management systems usually follow a binary classification approach to identify a peer as either good or bad. Few studies [14,15,38] have considered the multiclassification of peers. However, these works have considered neither predicting the specific peer model nor identifying other members of a malicious group. Therefore, this paper proposes Trutect to bridge these gaps.

## 3. System Design

As shown in Figure 1, the Trutect trust management system consists of three main components that can run on the cloud or in a fog gateway device. Alternatively, each component can be placed in a separate fog gateway device. However, in this case, these deceives are better placed in close vicinity to each other to reduce communication delays and possible communication problems.
Registry manager: The system registry manager is a centralized component responsible for administering and updating three lists: resource, transaction, and rating lists. The resource list maintains resources in the system and information on their owners. The transaction list contains the resource requests with the receiver and provider of each. Finally, the rating list includes the sent and received sublists for each peer. The sent list of a peer logs for each peer sending transactions, the resource sent, the resource receiver ID and the rating received for the transaction. The received list of a peer stores information about transactions where this peer was the resource receiver, including the provider, and the peer ratings of the transaction. The system registry manager updates the rating list after each transaction.NN component: This component employs an NN classifier to learn peers’ models (see Appendix A). It has twenty-five input nodes and five output nodes. The network weight is one and its depth is two as depicted in Figure 2. The NN is trained offline on a trust dataset that was constructed by simulating a P2P network using QTM [11]. The model simulated 100,000 transactions over 100,000 resources owned by 5000 peers. The peers included 1000 good peers (of whom 5% were pretrusted peers), 1000 pure malicious peers, 1000 feedback-skewing peers, 1000 malignant peers, and 1000 disguised malicious peers. Fifty percent of the malicious peers of each type are isolated, and 50% are in groups.Predictor: The predictor contains two main parts: the peer model predictor and the collective behavior analyzer. The peer model predictor feeds the provider information, based on the sent and received lists, into the trained NN, which predicts the peer model. If the peer model is good, then the requester peer is signaled to approve the peer as a candidate provider. Otherwise, if the peer is not an isolated malicious, the peer model is studied by the collective behavior analyzer to identify their group members. Finally, all transaction information is logged in the system registry and sent to the system administrator upon request.

The Trutect logic flow is diagrammed in Figure 3. Once a resource request arrives, the system randomly selects a provider of this resource from the resource list. Then, the NN predicter module determines, based on the sent and received lists of the provider, whether it is a good, purely malicious, feedback-skewing, malignant or disguised malicious peer. If the peer is good, the resource provider is approved, and the receiver rates the interaction as either positive or negative. Otherwise, if the receiver is malicious, the peer model is analyzed by the collective behavior analyzer, as shown in Figure 4, to determine which strategy the malicious peer is following, whether isolated or collective; and its group members, if any. Accordingly, the peer’s previous interactions and ratings are analyzed based on the sent and received lists; in addition, the generated information, which includes the malicious peer strategy and detected peer members, is sent to the system administrator to take an action and decide whether to proceed with resource sharing, provide a warning, or look for another provider.

## 4. Evaluation Methodology

A strictly controlled empirical evaluation framework was followed to evaluate the proposed algorithms. To allow for full control of the experimental parameters, we employed the renowned open-source simulator QTM [11]. The simulator imitates an assortment of network configurations and malicious peers’ behavioral models.

A P2P resource-sharing application was considered, although the system can be easily applied to other P2P applications. The following four context parameters were controlled to simulate representative samples of a P2P network:Percentages of malicious peers: Five different percentages of malicious peers were studied to test the robustness of the system: 15%, 30%, 45%, 60%, and 75%.Number of transactions: To simulate system performance under different loads, five values of the number of transactions were examined: 1000, 1500, 2000, 2500, and 3000 transactions.Malicious peer model: Four different types of malicious behavior were imitated, including pure, providers, feedback, and disguised, in each scenario to embody a real environment.Malicious strategies: Two malicious strategies, collective and isolated, were implemented in each scenario to represent real environments.

The number of peers and number of resources were fixed at 256 and 1000, respectively. Pretrusted peers were 5% of the total peers. For collective malicious groups, the number and size of groups were randomized. For simplicity, we considered a “closed world” network model where the peers within a network are static; they do not join and leave the network.

Two state-of-the-art algorithms, namely, EigenTrust [24] and InterTrust [25], and a reference case with no trust algorithm (none) were employed to benchmark the proposed algorithm performance.

Three performance metrics are used to assess the algorithm performance, the success rate, the running time, and the accuracy, as described below.
The success rate is represented in Equation (1) as the number of good resources received by good peers over the number of transactions attempted by good peers:(1)$$\mathrm{Success}\;\mathrm{rate}=\frac{\#\;\mathrm{of}\;\mathrm{good}\;\mathrm{resources}\;\mathrm{received}\;\mathrm{by}\;\mathrm{good}\;\mathrm{peers}}{\#\;\mathrm{of}\;\mathrm{transactions}\;\mathrm{attempted}\;\mathrm{by}\;\mathrm{good}\;\mathrm{peers}}$$The running time is defined as the total time of the algorithm’s execution in seconds. It is the time from when the system calls the main function of an algorithm until the control returns to the caller. Due to the sensitivity of the running time measure, all of the experiments were conducted on the same computer with an Intel core i7 CPU with a 1.1 GHz speed, 2 GB of RAM and a 200 GB hard disk.The classification accuracy is represented by the number of correctly classified sample cases over the number of all sample cases, as shown in Equation (2):(2)$$\mathrm{Classification}\;\mathrm{accuracy}=\frac{\#\;\mathrm{of}\;\mathrm{correctly}\;\mathrm{classified}\;\mathrm{cases}}{\#\;\mathrm{of}\;\mathrm{all}\;\mathrm{cases}\;}$$

## 5. Results and Discussion

This section describes the experimental results in terms of the success rate, running time, and accuracy. Each experiment was run several times, and the mean was calculated and analyzed.

### 5.1. Success Rate

The success rate can be calculated as the total number of good resources received by good peers divided by the total number of resources received by good peers, as in Equation (1).

Success rate is presented in Table 1, against the percentage of malicious peers as the number of transactions increases from 500 to 3000. The results suggest a correlation between increasing the percentage of malicious peers and the success rate of good peers when Trutect is utilized. This can be attributed to having more malicious peers involved in transactions, which highlights their malicious acts. However, this does not necessarily imply that increasing transactions would help in detecting more malicious peers. This is because the number of transactions will grow for both good and malicious peers. Hence, there will be more risks of good peers receiving bad resources or dishonest feedback. In addition, as transactions increase, there will be more confusion between some models of malicious peers, especially the disguised model and a mistaken good peer. As a result, more good peers may be classified as malicious peers and vice versa. However, increased interactions among malicious peers had adverse effects on EigenTrust, InterTrust, and the reference case (none). As shown in Figure 5, Trutect was notably superior to InterTrust, EigenTrust, and obviously the reference case in the success rate measure.

### 5.2. Running Time

Running time is defined as the total time of the algorithm’s execution in seconds. The running time is displayed against the percentage of malicious peers as the number of transactions increases from 500 to 3000 in Table 2.

As shown in the table, Trutect outperformed InterTrust in running time while simultaneously maintaining a high success rate. The running time for InterTrust remained constant for different percentages of malicious peers; in contrast, the running time for EigenTrust approximately doubled as the percentage of malicious peers increased. More importantly, in most cases, an increase in the malicious peer percentage decreased Trutect’s average running time. For example, the running time decreased by 21% as the malicious peer percentage increased from 15% to 75% when the transactions were 1000. The running time decreased as the percentage of malicious peers increased because the system did not always need to predict the peer type; the system had already learned the initial type from previous transactions. Increasing the number of transactions increased the running times for other algorithms, including the baseline scenario, none, that did not achieve meaningful results. In summary, as shown in Figure 6, In term of runtime, Trutect have a moderate and reasonable running time. Eigentrust was a good competitor with Trutect. However, Trutect is more efficient when it comes to busy and large networks. This can be attributed to the fact that most of the Trutect running time is due to neural network operations, while EigenTrust running time is dependent on the numbers of transactions and peers, so when these numbers grow largely, EigenTrust gets slower.

### 5.3. Accuracy in Predicting Malicious Model and Group Members

To illustrate Trutect’s ability to define a peer model and recognize its team members, screenshots from the system are shown in Figure 7. Given a peer ID, the system can define the peer type. It can also suggest peers that may be in the same malicious group. The examples show real peer models compared to predicted models and real and predicted members of groups. The following sections study this prediction accuracy in detail.

#### Predicting Malicious Models

An advantage of Trutect is its ability to determine peer models in addition to maliciousness. Some malicious types offer good resources but dishonest feedback; other types may offer bad resources but honest feedback. Ultimately, knowing specific behaviors will help users appropriately handle malicious peers.

The accuracies of detecting malicious models for isolated malicious peers are represented in Table 3. The accuracies are displayed against the percentage of malicious peers as the number of transactions increased from 500 to 3000.

Trutect successfully identified good, pure malicious, feedback-skewing malicious, and malignant provider peers and whether they were collective or isolated with an accuracy up to 97%. In detail, Trutect was able to predict pure malicious peers with an accuracy of up to 91% and no less than 71%. In the same way, the system could identify malignant providers with an accuracy of up to 97%, and no less than 70%. For disguised malicious peers, the highest and lowest accuracy values were 93% and 62%, respectively. For feedback-skewing malicious peer prediction, the accuracy had a positive correlation with transaction number: the accuracy increased alongside an increase in the number of transactions. The experiments showed that the system had some difficulty in identifying feedback malicious peers, since they act as good peers by providing good files. Table 4 first column sorts peer models based on the prediction accuracy of Trutect from high to low. Trutect was able to detect pure malicious peers with the highest accuracy value since they have obvious malicious behavior. Trutect faced some difficulties finding feedback types since they provide good files. On the other hand, malicious providers were easier to detect due to their bad files. Disguised malicios were more difficult to detect than pure and providers malicious because sometimes they behave as good peers which makes their true identity more confusing. It is interesting that a disguised malicious peer is easier to identify than a feedback malicious which may lead us to conclude that Trutect is more efficient when it comes to protect the network from malicious files than from dishonest feedback.

One contribution of Trutect is its ability to determine the other peers in a collective group. The accuracy of detecting the malicious group members of collective malicious peers is shown in Table 5 The accuracy is displayed against the percentage of malicious peers as the number of transactions increased from 500 to 3000.

The Trutect system’s accuracy of finding collective group members reached 91% and was no less than 87%. The accuracy remained almost stable, even when the number of transactions and the malicious peer percentage were changed. This situation occurred because a large part of the accuracy formula was specified for nonrelevant samples that were classified as nonrelevant. Although this scenario applies to the accuracy of defining collective and isolated malicious peers, it was clearer with the accuracy of the members because, in general, the number of team members for each peer will be less than the number of collective peers.

Table 4 second column sorts peer models based on the ability of Trutect to predict their group members from high to low. When finding members of a malicious team, we can see that pure malicious peers are also the easiest to identify due to their clear malicious behavior pattern. Furthermore, feedback malicious teams are easier to be identified than isolated counterparts due to their honest feedback to their teammates. The same goes for the malicious provider collectives who give their malicious group members good files while the bad files go to the peers outside their team. Finally, disguised malicious team members are the hardest to reveal since they may give their team members bad files or bad ratings in order to make their true identity hidden.

## 6. Conclusions

Many studies have considered discerning malicious peers, but none have focused on identifying the specific type and group members of a malicious peer. Trutect is proposed in this paper to bridge this gap. It exploits the power of neural networks to build models of each peer and classify them based on their behavior and communication patterns.

Trutect performance was evaluated against existing rival trust management systems considering the success rate, running time, and accuracy. For success rate, Trutect showed the highest success rate as it does not only reveal malicious peers but also their team members which makes it easier to identify good providers in the whole network after few rounds of transactions. Trutect also shows significant improvement in running time as the malicious peer percentage increases. Generally, Trutect recorded a significant result in identifying malicious peer models and group members. In addition, an increased number of transactions and a higher malicious peer percentage had clear positive effects on the accuracy of defining collective members in particular. In summary, Trutect is effective, especially on poisoned and busy networks with large percentages of malicious peers and enormous numbers of transactions.

Several interesting future research directions are opened by this research. For example, the accuracy of identifying disguised malicious peers might be improved by constructing a separate dataset with only good and disguised malicious peers. Additionally, Trutect’s performance at detecting malicious peers can be improved by considering dynamic learning where the system incrementally learns from newly classified instances. We would also consider sybil attackers who constantly change their identity to erase their bad histories.

## Figures and Tables

**Figure 1 sensors-21-04484-f001:**
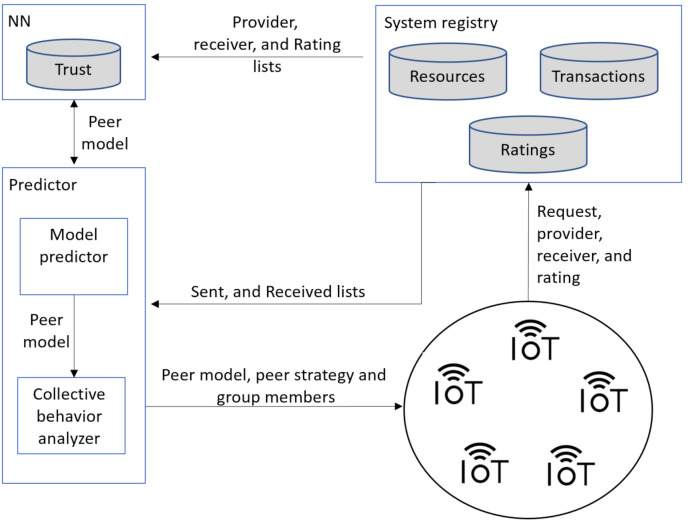
Trutect system architecture showing the main system components.

**Figure 2 sensors-21-04484-f002:**
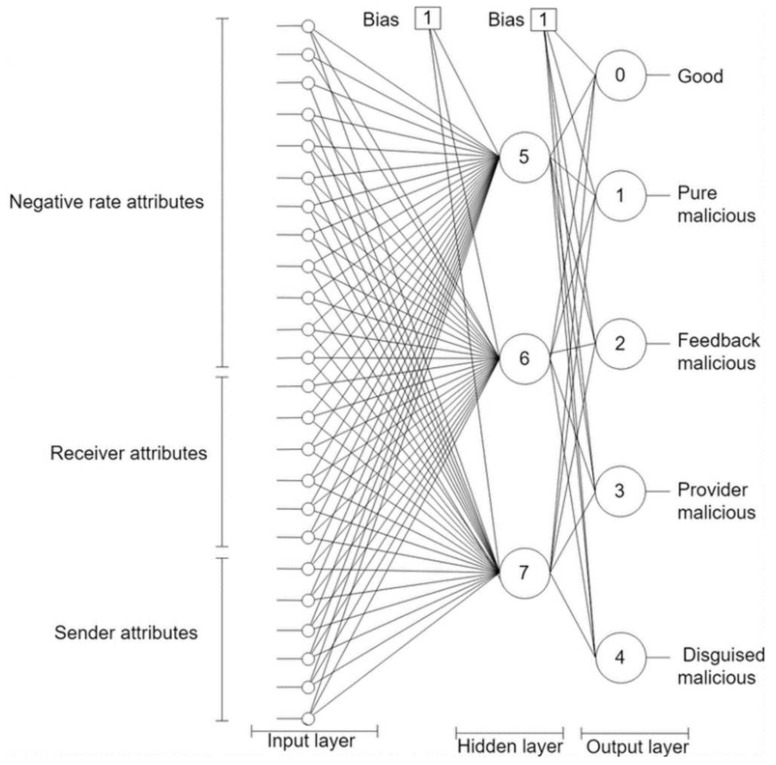
Trutect neural networks architecture to model peer behaviours.

**Figure 3 sensors-21-04484-f003:**
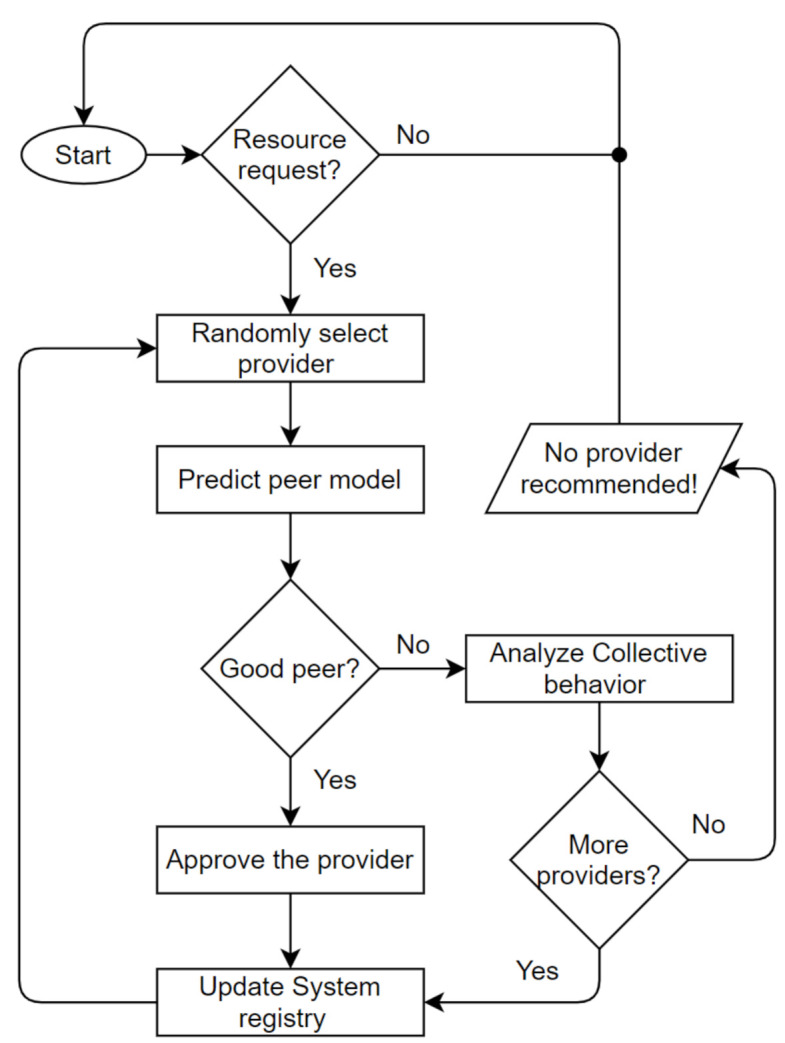
Trutect algorithm.

**Figure 4 sensors-21-04484-f004:**
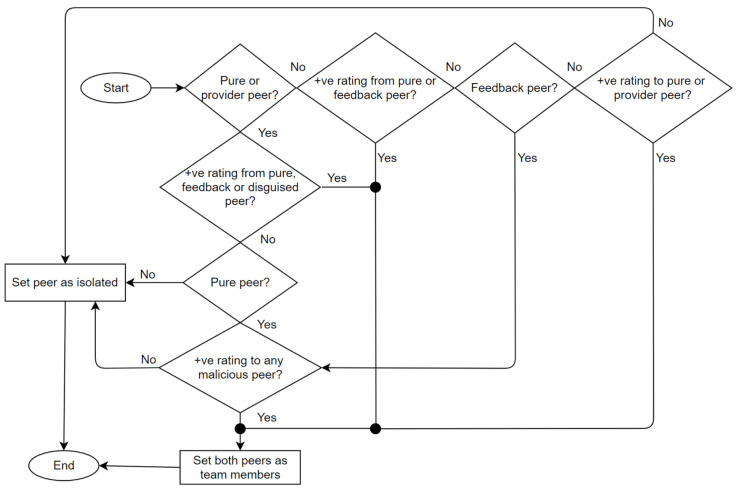
Collective behavior analyzer algorithm.

**Figure 5 sensors-21-04484-f005:**
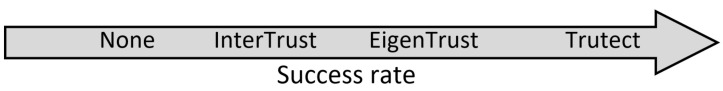
Algorithms in ascending order of success rate.

**Figure 6 sensors-21-04484-f006:**
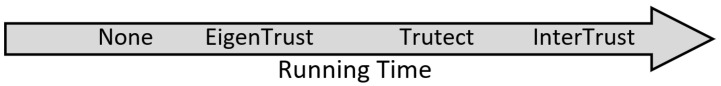
Algorithms in ascending order of running time.

**Figure 7 sensors-21-04484-f007:**
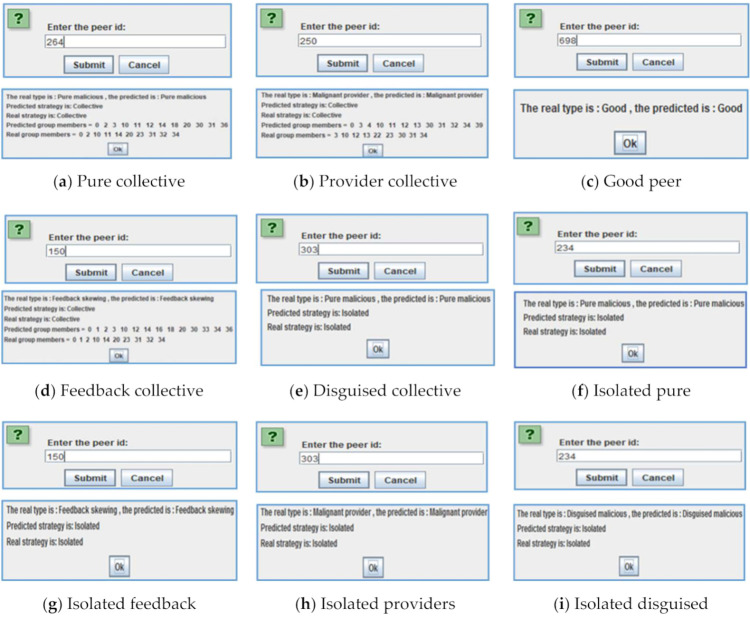
Screenshots of peer identification by Trutect.

**Table 1 sensors-21-04484-t001:** Success rate results as the percentage of malicious peers increases.

500 Transaction					2000 Transaction				
%Malicious	Trutect	EigenTrust	InterTrust	None	%Malicious	Trutect	EigenTrust	InterTrust	None
15	96.8	91.6	93.5	88.3	15	94.8	94.5	95.0	88.0
30	95.8	92.7	94.1	83.7	30	95.4	92.9	91.9	79.2
45	95.9	91.7	93.2	71.7	45	95.3	92.1	91.5	73.0
60	96.4	89.8	91.7	64.7	60	95.4	92.0	91.2	65.4
75	97.1	91.0	91.0	56.9	75	96.9	92.4	93.0	58.6
1000 transaction					2500 transaction				
%Malicious	Trutect	EigenTrust	InterTrust	None	%Malicious	Trutect	EigenTrust	InterTrust	None
15	94.7	93.5	93.5	87.4	15	95.1	94.7	94.4	89.2
30	96.0	94.3	92.3	83.0	30	95.0	94.8	93.7	80.0
45	96.1	90.7	90.5	75.8	45	95.8	90.4	91.6	74.5
60	94.9	92.9	92.6	63.	60	95.9	94.2	92.2	65.3
75	97.7	91.0	91.0	56.4	75	97.0	92.5	93	57.5
1500 transaction					3000 transaction				
%Malicious	Trutect	EigenTrust	InterTrust	None	%Malicious	Trutect	EigenTrust	InterTrust	None
15	95.3	94.7	93.3	88.0	15	95.0	94.9	94.6	89.6
30	95.3	93.5	92.9	80.2	30	94.7	92.8	93.4	80.7
45	96.7	93.3	92.1	75.0	45	95.2	92.6	93.6	75.4
60	97.0	91.9	92.1	66.8	60	95.6	92.9	92.0	64.0
75	96.9	91.9	92.2	60.3	75	97.3	90.8	90.9	57.0

**Table 2 sensors-21-04484-t002:** Run time results as the percentage of malicious peers increseas.

500 Transaction					2000 Transaction				
%Malicious	Trutect	EigenTrust	InterTrust	None	%Malicious	Trutect	EigenTrust	InterTrust	None
15	498	2	465	0.97	15	877	8.38	1348	10
30	547	6	446	0.95	30	1123	54	1337	2.22
45	502	27	446	2.02	45	959	150	1344	2.55
60	497	71	461	1.13	60	938	207	1488	2.23
75	452	229	463	1.12	75	915	337	1421	0.76
1000 transaction					2500 transaction				
%Malicious	Trutect	EigenTrust	InterTrust	None	%Malicious	Trutect	EigenTrust	InterTrust	None
15	1006	5	1236	1.23	15	889	20	1801	0.48
30	1000	26	1144	1.39	30	1001	121	1833.	1.41
45	1235	109	1024	0.44	45	844	188	1833	2.23
60	758	152	920	0.55	60	814	359	1807	1.90
75	793	263	1060	0.63	75	751	488	1833	2.03
1500 transaction					3000 transaction				
%Malicious	Trutect	EigenTrust	InterTrust	None	%Malicious	Trutect	EigenTrust	InterTrust	None
15	889	11	1801	8.87	15	903	31	3327	0.57
30	1152	72	1833	1.15	30	866	157	3036	0.65
45	1170	180	1726	1.28	45	821	378	3038	0.75
60	786	321	1537	3.29	60	772	405	3021	0.95
75	730	401	1436	1.81	75	813	508	2363	2.00

**Table 3 sensors-21-04484-t003:** Accuracy of predicting peer models.

500 Transaction					2000 Transaction				
%Malicious	Pure	Feedback	Providers	Disguised	%Malicious	Pure	Feedback	Providers	Disguised
15	0.85	0.83	0.83	0.86	15	0.8	0.81	0.97	0.92
30	0.77	0.52	0.78	0.83	30	0.81	0.76	0.81	0.85
45	0.88	0.71	0.84	0.89	45	0.8	0.77	0.80	0.9
60	0.78	0.78	0.838	0.79	60	0.72	0.7	0.80	0.73
75	0.85	0.78	0.90	0.84	75	0.8	0.7	0.80	0.67
1000 transaction					2500 transaction				
%Malicious	Pure	Feedback	Providers	Disguised	%Malicious	Pure	Feedback	Providers	Disguised
15	0.90	0.85	0.92	0.83	15	0.8	0.81	0.75	0.72
30	0.84	0.79	0.80	0.75	30	0.76	0.77	0.8	0.61
45	0.8	0.73	0.87	0.66	45	0.77	0.75	0.83	0.80
60	0.75	0.70	0.83	0.88	60	0.77	0.71	0.77	0.70
75	0.73	0.70	0.79	0.84	75	0.75	0.65	0.72	0.85
1500 transaction					3000 transaction				
%Malicious	Pure	Feedback	Providers	Disguised	%Malicious	Pure	Feedback	Providers	Disguised
15	0.91	0.91	0.90	0.93	15	0.80	0.90	0.88	0.80
30	0.87	0.92	0.87	0.91	30	0.80	0.76	0.84	0.82
45	0.75	0.75	0.81	0.79	45	0.80	0.70	0.70	0.81
60	0.76	0.77	0.79	0.80	60	0.72	0.70	0.80	0.7
75	0.75	0.70	0.80	0.76	75	0.71	0.62	0.85	0.70

**Table 4 sensors-21-04484-t004:** Peer model and group members in descending order of prediction accuracy by Trutect.

	Peer Model Prediction Accuracy	Group Members Prediction Accuracy
Highest	Pure	Pure
	Providers	Feedback
	Disguised	Providers
Lowest	Feedback	Disguised

**Table 5 sensors-21-04484-t005:** Accuracy in predicting Group Members.

500					2000				
%Malicious	Pure	Feedback	Providers	Disguised	%Malicious	Pure	Feedback	Providers	Disguised
15	0.93	0.93	0.94	0.94	15	0.97	0.92	0.95	0.95
30	0.90	0.90	0.88	0.91	30	0.93	0.92	0.9	0.92
45	0.92	0.93	0.94	0.94	45	0.92	0.93	0.92	0.9
60	0.92	0.92	0.92	0.92	60	0.924	0.93	0.93	0.92
75	0.93	0.95	0.94	0.93	75	0.94	0.92	0.92	0.9
1000					2500				
%Malicious	Pure	Feedback	Providers	Disguised	%Malicious	Pure	Feedback	Providers	Disguised
15	0.95	0.93	0.94	0.92	15	0.95	0.94	0.93	0.93
30	0.92	0.91	0.94	0.92	30	0.95	0.94	0.95	0.94
45	0.96	0.96	0.95	0.94	45	0.91	0.9	0.9	0.87
60	0.9	0.9	0.91	0.91	60	0.93	0.93	0.92	0.91
75	0.93	0.92	0.92	0.91	75	0.95	0.95	0.9	0.9
1500					3000				
%Malicious	Pure	Feedback	Providers	Disguised	%Malicious	Pure	Feedback	Providers	Disguised
15	0.95	0.94	0.95	0.94	15	0.93	0.94	0.95	0.91
30	0.95	0.94	0.95	0.95	30	0.94	0.95	0.95	0.94
45	0.93	0.91	0.92	0.9	45	0.94	0.95	0.95	0.94
60	0.9	0.9	0.9	0.9	60	0.94	0.92	0.93	0.89
75	0.92	0.92	0.9	0.9	75	0.97	0.95	0.95	0.95

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
