# Peer review of "Managing Trust and Detecting Malicious Groups in Peer-to-Peer IoT Networks"

_sensors, 2021, doi:10.3390/s21134484_

Round 1
Reviewer 1 Report
In the context of the emerging Internet of Things (IOT), this paper studies the problem of identifying various profiles of "malicious" devices based on feedback from resource-sharing "transactions" between devices.
The premise of the work is that IOT devices conduct "transactions" where they request and receive resources from other devices in a peer-to-peer (P2P) manner. The traces are followed by some sort of feedback. Among the devices, some are "good" and other are "malicious" and aim to disrupt the network. These malicious nodes in the network match one of four different profiles: "pure malicious", "camouflaged malicious", "feedback-skewing", and "malignant".
The paper reports on experiments where a neural network is used to identify the "type" of a peer based on a collection of feedback, and the overall performance of the trust management system is evaluated by inserting the "malicious peer" detection / trust evaluation in the peer selection process before resources are requested.
The system is compared with the well-known Eigentrust algorithm and the "Inter-trust" algorithm.
I have several fundamental concerns over this paper.
First of all, it's unclear where in an IOT deployment this would work. Does the smart thermostat request resources from the smart refrigerator or the smart doorbell, which then turns out to be malicious? It seems to me that in most deployments proximity sharing would happen within a network of devices owned by a single entity. And the feedback mechanism is entirely hypothetical, it seems. Most trust management work places itself in the context of P2P transactions over the internet, which are then rated by people (mostly). Generally, I think the context and motivation needs to be much clearer and convincing.
Secondly, the state of the art in terms of reputation management systems is not very deeply covered. I would not expect this paper to survey all the possible trust management systems, but either link to a survey or focus on those that have been studied in the context of IOT (which would also help provide some justification for the chosen context, see point above).
I also think there is a dimension missing in the analysis: the authors suggest that trust is all about detecting malicious peers in a binary manner, but originally trust is subjective: A trusts B, but C might not trust B. Trust management systems might not attempt to decide whether B is "good" or "bad", but rather, say, whether some peer D should trust B. The result might depend on D's position with respect to A and C. Finally, are Eigentrust and InterTrust really "state of the art"?
Third, I have an issue with the main idea of the paper -- attempting to identify specific peer profiles via behavioural traces and machine learning -- is a bit shaky. While the four listed profiles are reasonable threat models to use in the evaluation of a reputation system, they are still only models, and as far as I know there is no evidence that attackers would precisely follow one of these models. So I don't think it's reasonable to use the model as an input of the defense system, and attempt to recognize these exact models. What would happen in the real world, if an attacker slightly deviated from the models described? Would the defense crumble entirely?
Finally, a less subjective but crucial point is that the description of the method (for the above problem of identifying peer profiles) is very poor: all we know that it's done with a 2-layer neural net, with 25 input nodes and 5 output nodes. There is no indication of the type or depth of the network, how the feature vectors were built from interaction traces... basically this work cannot be reproduced.
The evaluation metrics are reasonable, but the exact figures are not given nor are they evaluated for statistical significance. It looks like in many cases accuracy difference is less than one percentage point, which may not be significant.
Overall, I think the most problematic point is the lack of technical description for the neural network and its training process. I would also be interested to see what would happen if the peer profiles in the evaluation differed from those in the training phase (which is bound to be the case in the real world).
Reviewer 2 Report
General Considerations:
This paper proposes a trust management system that exploits the power of neural networks to identify the specific peer model and its group members. A large dataset of behavioral models in P2P networks can be used to advance research in the field. And a well-controlled evaluation framework to study the performance of a trust management system.
The authors aim the usage of neural networks in this study, but in the section of related works, it is just compared with classical machine learning algorithms, such as decision trees, and KNN, and it represents a lack of the study due to the different behaviors of these algorithms. Just reference 14 is related to the proposed paper.
As for the association of the proposed objective in the work about its conclusion, it is necessary to improve, focusing on the proposed theme and with results obtained in a more consolidated way.
There is a lack of consistency in the results (the work has many graphs, some unnecessary and there is an improvement in the exploitation of the results), it has many graphs with few explanations, needing to explore the architecture better.
The article talks about the Management of trust and detection of malicious groups in IoT networks P2P, however, I believe that there was a lack of referencing similar works related to the topic of trust in distributed systems. For example, the following articles. (I suggest you read these articles)
[1] Adnane, Asma, Christophe Bidan, and Rafael Timóteo de Sousa Júnior. "Trust-based security for the OLSR routing protocol." Computer Communications 36.10-11 (2013): 1159-1171.
[2]“Canedo, Edna Dias, Rafael Timóteo de Sousa Junior, and Robson de Oliveira Albuquerque. "Trust model for reliable file exchange in cloud computing." International Journal of Computer Science & Information Technology 4.1 (2012): 1.
[3] de Oliveira Albuquerque, Robson, et al. "Leveraging information security and computational trust for cybersecurity." The Journal of Supercomputing 72.10 (2016): 3729-3763.
[4] Canedo, Edna Dias, Robson de Oliveira Albuquerque, and Rafael Timóteo de Sousa Junior. "Trust model for file sharing in cloud computing." The Second International Conference on Cloud Computing, GRIDs, and Virtualization. Rome, Italy: ACM, 2011.
Structural problems:
The captions of the Figures must be auto-explained.
The charts of the measurements should be a table where can reduce a lot of space and your text can be fluid and avoid the unnecessary use of the pages.
Instead of the box of the proposed system (Fig. 6), the authors should draw a block diagram of the system components.
There are some typos in the text and some grammar mistakes, e.g in Equation 2.
Figures 4, 5, 6 have hard read captions and the colors of the legend are not equal with which that are presented. I recommend keeping the pattern. Due to the usage of different algorithms, this kind of chart doesn’t allow a good read of the data, so a table that compares the results of the solutions is much interesting.
Round 2
Reviewer 1 Report
This updated version of the paper addressed the main issue I had raised, which was the lack of technical description. The diagram presenting the structure of the network is a valuable addition. However, I would like to see some more details about the input features. If space is lacking, they should come in an appendix. As it is, it is still difficult to reproduce the work, and reproducibility is essential for scientific publication.
With the more detailed results (tables), I also have more questions regarding the numeric results.
It seems that for the traditional trust algorithms, performance decreases with the number of malicious peers and increases with the number of transactions. The "trutect" system, on the other hand, sees the opposite: performance increases with the number of malicious peers and decreases with the number of transactions. Out of curiosity, I plotted the performance vs. number of transactions and vs. percentage of malicious peers (averaging performance across the other dimension) and the trends are very clear. The text includes one sentence explaining that performance increases with the number of malicious peers, explaining that it allows for better detection, but this contradicts the other result (performance decreasing with number of transactions).
It seems to me that this should be explained or at least analyzed in more depth -- I realize that neural networks' behaviors can be difficult to explain, but certainly it's a puzzling result that should be scrutinized.
Finally I would mention that the additional paragraph at the beginning is absolutely not convincing as to the relevance of the proposed system. The example is the case of a smart lock that could lock a person out of their house or let in a stranger, because an identity verification mechanism was compromised. In this case, I don't see where the transaction is: which device is the resource provider, what is the resource, and which is the recipient? Does the person's phone act as the recipient of a conceptual resource (the house getting locked) and ask the person to check whether he/she is satisfied of the house being locked? It seems to me that a single "bad transaction" here has potentially huge consequences and one cannot let the system run with hundreds of transactions before looking into it.
I looked into one of the surveys added as a reference and I found a paper discussing trust issues in an IOT network: it seems to be about routing security, which could possibly be framed as "resources being provided", but in this case one difficulty is that the sender of a packet cannot tell wether the next node is malicious or whether routing failed further down the road.
At the end of the day this is not a complete deal-breaker for the paper; but certainly it would be nice to see an application scenario where the scheme makes sense.
Reviewer 2 Report
The theme of the work has great academic relevance, the corrections were very relevant to improve the work, after the corrections the objective of the work can be clearly observed.
As far as concerns my considerations were met. However, it is necessary to carry out a general reading of the work in order to validate the other considerations focusing on the state of the art, objectives and conclusion
Author Response
Thank you for your reading of the manuscript and for the constructive comments. We have taken the comments into consideration to improve the quality of the manuscript.
